# Efficacy of *Bacillus subtilis* ANSB060 Biodegradation Product for the Reduction of the Milk Aflatoxin M_1_ Content of Dairy Cows Exposed to Aflatoxin B_1_

**DOI:** 10.3390/toxins11030161

**Published:** 2019-03-13

**Authors:** Yongpeng Guo, Yong Zhang, Chen Wei, Qiugang Ma, Cheng Ji, Jianyun Zhang, Lihong Zhao

**Affiliations:** 1State Key Laboratory of Animal Nutrition, College of Animal Science and Technology, China Agricultural University, Beijing 100193, China; 18771951786@163.com (Y.G.); maqiugang@cau.edu.cn (Q.M.); jicheng@cau.edu.cn (C.J.); jyzhang@cau.edu.cn (J.Z.); 2College of Biological Engineering, Henan University of Technology, Zhengzhou 450001, China; yongzhang208@163.com; 3Yili Vocational and Technical College, Yining 835000, China; 1468472746@163.com

**Keywords:** *Bacillus subtilis* ANSB060, aflatoxin B_1_, aflatoxin M_1_, milk, dairy cows

## Abstract

This study was conducted to determine the effect of *Bacillus subtilis* ANSB060 biodegradation product (BDP) in reducing the milk aflatoxin M_1_ (AFM_1_) content of dairy cows fed a diet contaminated with aflatoxin B_1_ (AFB_1_). Twenty-four Chinese Holstein cows (254 ± 19 d in milk; milk production 19.0 ± 1.2 kg d^−1^) were assigned to three dietary treatments, as follows: (1) control diet (CON), consisting of a basal total mixed ration (TMR); (2) aflatoxin diet (AF), containing CON plus 63 μg of AFB_1_ kg^−1^ of diet dry matter; and (3) aflatoxin diet plus BDP (AF + BDP), containing AF plus BDP at 0.2% of diet dry matter. The experiment lasted 12 days, including an AFB_1_-dosing period from days one to eight, followed by a clearance period from days nine to twelve. Milk samples were collected on days 2, 4, 6, and 8–12, and the plasma was sampled on day 9, before morning feeding. Short-term AFB_1_ exposure did not affect the milk production and composition. The plasma biochemical indices, except for lactic dehydrogenase (LDH), were also not changed by the AFB_1_ intake. The plasma LDH level was significantly elevated (*p* < 0.05) following dietary treatment with AFB_1_, while no significant difference was observed between the AF + BDP and CON treatments. Adding BDP to the AFB_1_-contaminaed diet resulted in a significant reduction in AFM_1_ concentration (483 vs. 665 ng L^−1^) in the milk, AFM_1_ excretion (9.14 vs. 12.71 μg d^−1^), and transfer rate of dietary AFB_1_ to milk AFM_1_ (0.76 vs. 1.06%). In conclusion, the addition of BDP could be an alternative method for reducing the dietary AFB_1_ bioavailability in dairy cows.

## 1. Introduction

Aflatoxins (AF) are harmful secondary metabolites mainly produced by *Aspergillus flavus* and *Aspergillus parasiticus* fungi. Among the eighteen different types of aflatoxins, the major naturally-occurring members are aflatoxin B_1_, B_2_, G_1_, and G_2_. AFB_1_ is the most prevalent and toxic, and has been classified as a Group I human carcinogen by the International Agency for Research on Cancer (IARC). After ingestion by livestock animals, AFB_1_ is partly bio-transformed into aflatoxin M_1_ (AFM_1_) in the liver by the mitochondrial cytochrome P450 oxidative system, which is then secreted into the milk of lactating animals, including dairy cattle. The carcinogenicity of AFM_1_ is about 10 times lower than that of AFB_1_; however, unlike AFB_1_, AFM_1_ exerts a direct cytotoxicity on human cells in the absence of metabolic activation. The transfer rate of dietary AFB_1_ to milk AFM_1_ mainly depends on the milk yield, and is usually 1%–2% for low-yielding cows (30 kg milk yield per day) and up to ~6% for high-yielding cows (>30 kg milk yield per day) [1]. Milk contamination with AFM_1_ has attracted worldwide attention because of the high consumption of milk and dairy products by humans, especially children. Considering the health risks associated with the human dietary exposure to AFM_1_, more than 60 countries have set strict guidelines for maximum residue level (MRL) of AFM_1_ in milk [2]. While in the United States and China, the maximum allowable concentration of AFM_1_ in fluid milk is 0.5 μg L^–1^, the legal limit is much more stringent in the European Union, where the level is set at 0.05 μg L^−1^. To avoid carry-over, the maximum permissible amount of AFB_1_ in dairy feed has also been established, ranging from 20 μg kg^−1^ in the United States, to 10 μg kg^−1^ in China, and 5 μg kg^−1^ in the European Union.

The pre-harvest prevention of aflatoxins occurrence and the post-harvest elimination of contamination are the main strategies to reduce aflatoxicosis in human and animals [3]. The application of good agricultural practices (GAPs), such as crop rotation, harvesting at the right time, control of insect damage, and choice of fungal resistant varieties, is helpful for inhibiting fungal growth and aflatoxins production. Meanwhile, strategies for post-harvest decontamination include physical, chemical, or biological methods. Physical treatments like thermal inactivation, irradiation, and extrusion generally do not comply with the cost and productivity requirements for commercialization [4]. The addition of mycotoxin binders to contaminated diets is also a physical method, which has been widely applied to reduce AFB_1_ absorption in dairy cows. Common types of mycotoxin binders include calcium montmorillonite clay [5], aluminosilicate clay [6], and yeast cell culture [7]. However, some of these adsorbents may also bind minerals, vitamins, and amino acids in feeds [8], as well as reducing the efficiency of the pharmacokinetics of antibiotics [9]. The use of chemical methods comprising ammoniation, ozonation, and peroxidation in food and feeds is limited as a result of the potential toxicity of chemical residues [10]. The biological degradation of aflatoxins, using microorganisms and enzymes, has been considered to be a promising strategy for lessening the negative effects of dietary AFB_1_ in animals. The major advantage of biological detoxification methods is that the enzymes that come from microbes can transform aflatoxins into non-toxic or less toxic metabolites under mild conditions, with a minor impact on the palatability and nutritive quality of food and feeds [11].

Some species of microbes, including fungal and bacterial strains, such as *Armillariella tabescens* [12], *Pleurotus pulmonarius* [13], *Rhodococcus erythropolis* [14], *Bacillus subtilis* [15], and *Bacillus licheniformis* [16], have been reported to biodegrade AFB_1_ in vitro. However, few studies have evaluated their detoxification efficiency in vivo. *B. subtilis* ANSB060 was isolated from fish gut, which can degrade AFB_1_, AFM_1_, and AFG_1_ by 81.5%, 60%, and 80.7%, respectively, in liquid culture [15]. In addition, *B. subtilis* ANSB060 has been shown to effectively alleviate aflatoxicosis in layers [17], broilers [18], ducks [19], and carps [20]. The dietary supplementation of a *B. subtilis* ANSB060 fermentation product can also reduce the accumulation of AFB_1_ and AFM_1_ in the liver of birds, suggesting that *B. subtilis* ANSB060 can degrade aflatoxins in animal gastrointestinal tracts. *B. subtilis* is a transient microorganism of the digestive tract, and the bacteria is capable of forming spores resistant to heat and a low gastric pH. Furthermore, *B. subtilis* is generally recognized as safe (GRAS) status for nutritional and pharmaceutical use. Spores of the *Bacillus* species have been used as direct-fed microbial feed additives for ruminants [21,22]. Thus, the aim of this study was to investigate the effectiveness of the *B. subtilis* ANSB060 biodegradation product (BDP) in reducing the AFM_1_ content in the milk of dairy cows exposed to AFB_1_.

## 2. Results and Discussion

### 2.1. Production Performances

Over the 12-day feeding trial, all of the cows in the three treatments behaved normally, without observed clinical signs of aflatoxicosis. As shown in Table 1, there were no treatment differences on the milk production and components (milk protein, fat, lactose, milk urea nitrogen (MUN), and somatic cell count (SCC)) of the dairy cows. In accordance with the results of our study, the report of Kutz et al. [23] showed that the incorporation of 112 µg kg^−1^ of AFB_1_ to the diet did not affect the feed intake, milk production, and component concentrations of dairy cows. Consistently, Maki et al. [5] and Sulzberger et al. [24] also found no changes in the milk yield and composition of dairy cows in early to mid-lactation when administrated with AFB_1_ at doses of 79 and 100 µg kg^−1^ in the feed. However, Queiroz et al. [25] reported that dosing the cows with AFB_1_ at 75 µg kg^−1^ of diet resulted in a significant reduction of milk protein concentration and milk fat yield. It should be noted that the cows were exposed to naturally contaminated diets, which might have contained multiple mycotoxins in the study of Queiroz et al. [25]. The co-exposure to the mycotoxin combinations may have led to more adverse effects on the cows than purified AFB_1_, due to the possible additive or synergic effect.

### 2.2. Plasma Biochemical Indices

The effects of dosing AFB_1_ with or without BDP on the plasma parameters of dairy cows are presented in Table 2. The short-term addition of AFB_1_ to the diet did not cause statistically significant changes in the plasma biochemical indices tested in the current study, except for lactic dehydrogenase (LDH), an indicator widely used to evaluate the presence of cells and tissues damage. The plasma LDH level of the dairy cows was remarkably increased in the AF treatment when compared with that of the CON treatment. In contrast, there was no significant difference in the plasma LDH level between the CON and AF + BDP treatments. To our knowledge, few studies have investigated the impact of AFB_1_ exposure on the plasma LDH content of dairy cows. A contradictory result was present in the literature regarding the response of ewes to AFB_1_ challenge, in which the serum LDH level was not affected by the incorporation of AFB_1_ in the diet [26]. The liver is the main site of AFB_1_ metabolism, and a major target organ for aflatoxicosis. In agreement with the current study, the main indices of liver injury, like serum alkaline phosphatase (ALP), aspartate amino transferase (AST), and γ-glutamyl transferase (GGT), were not changed in the dairy cows following dietary exposure to 100 µg of AFB_1_ kg^−1^ of diet [27]. In addition, Xiong et al. [7,27] also found no effects of dietary AFB_1_ on the liver function parameters of dairy cows. Taken together, the elevated plasma LDH level in AF treatment might not be due to liver damage in this experiment, and further study is needed in order to understand the increased plasma LDH level in the dairy cows exposed to AFB_1_.

### 2.3. AFM_1_ Content in Milk 

The aflatoxins (AFB_1_, AFB_2_, AFG_1_, and AFG_2_) contents in the TMR were below the detection limits. Prior to dietary treatment with AFB_1_, AFM_1_ was not detected in the milk samples of all of the cows. Also, no AFM_1_ was found in the milk from the CON treatment during the experimental period. The patterns of milk AFM_1_ concentration in the AF and AF + BDP treatments are shown in Figure 1. Previous studies have confirmed that AFM_1_ can be found in the milk of dairy cows [1,28], ewes [29], and goats [30] within 24 h of the first AFB_1_ administration. In the present study, milk AFM_1_ reached a mean of 360 and 309 ng L^−1^ in the AF and AF + BDP treatments, respectively, on the second day of the AFB_1_-dosing period. The study of Moschini et al. [31] showed that AFB_1_ could be rapidly absorbed through the rumen wall before reaching the intestine, and could be transformed into AFM_1_ in the liver of dairy cows as soon as 15 minutes after a single oral intake of 4.9 mg AFB_1_. A plateau of AFM_1_ concentration in the milk was observed from day 4 of administration, and the steady state condition was maintained up to the last day of the AFB_1_-dosing period (day 8). When the AFB_1_ administration was withdrawn, the milk AFM_1_ concentration dropped rapidly and was below the Chinese legislative limit (0.5 μg L^–1^) within 24 h (day 9). The milk AFM_1_ was not detected by day 12 in both AF and AF + BDP treatments. The time of disappearance of milk AFM_1_ after the last dietary administration of AFB_1_ was in agreement with published data. Xiong et al. [7] reported that no AFM_1_ was detected in the milk of dairy cows after day 3 of the AFB_1_ withdrawal period, regardless of whether the level of dietary AFB_1_ was 20 or 40 μg kg^−1^. Moreover, Queiroz et al. [25] and Ogunade et al. [28] demonstrated that AFM_1_ cleared from the milk of the dairy cows fed a diet contaminated with AFB_1_ (75 μg kg^−1^) 72 h after withdrawing the toxin. These findings suggested that the clearance of AFM_1_ from milk after the ending of AFB_1_ administration is not related to the dietary AFB_1_ content.

The effects of dosing AFB_1_ with or without BDP on the AFM_1_ content in the milk of dairy cows are summarized in Table 3. The average AFM_1_ concentration in milk at steady state was 665 ng L^−1^ in the AF treatment, 1.33 and 133 times higher than the maximum allowable level set by China (0.5 μg L^−1^) and the European Union (0.05 μg L^−1^), respectively, while the mean AFM_1_ concentration (483 ng L^−1^) in the AF + BDP treatment was below the Chinese legislative limit. The carryover rate of AFB_1_ from feed into AFM_1_ in milk was 1.06% in the AF treatment, and the value was in agreement with the results reported by Maki et al. [5] and Ogunade et al. [28], in which the observed transfer rate was 1.07% and 1.13%, in dairy cows challenged with 100 μg kg^−1^ and 75 μg kg^−1^ of AFB_1_ in their diet, respectively. However, Britzi et al. [1] showed that the carryover rate of AFB_1_ into AFM_1_ from high producing cows (milk yield of >35 kg per day) was on average 5.4%. Accumulating evidence has suggested that the milk yield is the main factor contributing to the variability of the transfer rate of feed AFB_1_ to milk AFM_1_ in dairy cows [32,33,34]. Britzi et al. [1] described the relationship between the carryover of AFB_1_ to AFM_1_ in milk and the milk yield as follows: carry-over% = 0.5154 e^0. 0521 × milk yield^ (*r*^2^ = 0. 6224). A linear regression equation was described was proposed by Masoero et al., as follows: carry-over% = −0.326 + 0.077 × milk yield; *r*^2^ = 0. 58) [32]. It is noteworthy that the model developed by Masoero et al. [32] fitted the data of the present study well, with the actual transfer rate and estimated transfer rate being 1.06% and 1.16%, respectively. 

The addition of BDP to the AFB_1_-contaminated diet resulted in a 27.4% reduction in milk AFM_1_ concentration, a 28.1% reduction in AFM_1_ excretion, and a 28.3% reduction in AFB_1_ transfer from feed to milk. Previous studies have reported that BDP reduced the bioavailability and toxicity of AFB_1_ in livestock animals. The addition of BPD (0.2%) to an aflatoxins-contaminated diet resulted in a 62.5% and 40.0% reduction of AFB_1_ and AFM_1_ residues in the liver of broilers, as well as a 76.7% and 79.9% reduction of AFB_1_ and AFB_2_ recovered from duodenal content [18,35]. More recently, Fan et al. [20] also found that incorporating BDP at the level of 0.1% in the diet significantly lowered the AFB_1_ residues by 94.4% and 92.2% in the hepatopancreas and gonad of Yellow River carps, respectively, during the period of exposure to AFB_1_. Previous studies have shown that *B. subtilis* ANSB060 degraded AFB_1_ by the extracellular constitutively produced enzyme, as the culture supernatant of ANSB060 was able to degrade 78.7% of AFB_1_ after incubation with the toxin for 72 h [15]. Although it was not clear whether the spores of ANSB060 could germinate in the gastrointestinal tract of dairy cows and whether they displayed an AFB_1_ degradation activity, the constitutive aflatoxins degrading enzymes in the ANSB060 fermentation product should be responsible for the AFB_1_ reduction. More research is required to explore this question.

## 3. Conclusions

Feed contamination with AFB_1_ is of great concern in the dairy industry, because of the inherent risk of AFM_1_ residues in milk and milk products intended for human consumption. The inclusion of AFB_1_ (63 µg kg^−1^) in the diet increased the milk AFM_1_ concentration beyond the Chinese action level of 0.5 µg L^−1^ in lactating dairy cows, under the current study conditions. The addition of *Bacillus subtilis* ANSB060 biodegradation product to the diet of the cows exposed to AFB_1_ resulted in a significant reduction in the milk AFM_1_ concentration (483 vs. 665 ng L^−1^), AFM_1_ excretion (9.14 vs.12.71 μg d^−1^), and AFB_1_ transfer rate (0.76 vs. 1.06%) from feed to milk. Additional studies are needed in order to investigate the underlying mechanisms involved in the AFB_1_ detoxification process of *B. subtilis* ANSB060, and the influences of AFB_1_ sequestration by BDP in vivo, aiming the commercial application of this product as a biological agent for AFM_1_ reduction.

## 4. Materials and Methods 

### 4.1. Animals, Experimental Design and Diets

The cows in this study were cared for according to the protocols approved by the Animal Welfare Committee of the China Agricultural University (ethical approval code: CAU20180630-2; Date: 30 June 2018). Twenty-four Holstein cows, averaging 254 ± 19 day in milk (DIM), were assigned to three groups balanced for milk production, body weight, and parity. Dietary treatments included the following: (1) control diet (CON), consisting of a basal total mixed ration (TMR); (2) aflatoxin diet (AF), containing CON plus 63 μg kg^−1^ of AFB_1_ of diet dry matter; (3) aflatoxin diet plus BDP (AF + BDP), containing AF plus BDP at 0.2% of diet dry matter. The BDP consisted of 80% spray-dried fermentation product of *B. subtilis* ANSB060 and 20% rice husk meal as the carrier. The viable spores of *B. subtilis* ANSB060 in the BDP were 2 × 10^9^ CFU g^−1^ by the plate count. The diets were formulated according to the nutrient requirements of the Holstein cows in late lactation producing 20 kg of milk d^−1^ [36]. The ingredient and chemical compositions of the basal TMR are shown in Table 4. The cows were fed AFB_1_ and BDP based on an estimated average of 19 kg of dry matter intake (DMI) each day, resulting in a daily intake of 1197 µg of AFB_1_ (19 kg of DMI × 63 µg of AFB1 kg^−1^) and 38 g (19 kg of DMI × 0.2%) of BDP. The 12-day trial was divided into an AFB_1_-dosing phase (days 1 to 8) and a clearing phase (days 9 to 12). The cows were fed the TMR in two equal aliquots daily (at 09:30 and 16:30). During the AFB_1_-dosing phase, the daily dose of AFB_1_ and BDP for each animal was divided into two aliquots. Each aliquot of AFB_1_ was mixed with ground corn, placed into a digestible gelatin capsule, and orally administered with the help of a balling gun. Each aliquot of BDP was mixed with 150 g of ground corn so as to encourage consumption. All of the cows in the three treatments were only fed the basal TMR during the following four-day clearance period. The contents of the mycotoxins (AFB_1_, AFB_2_, AFG_1_, AFG_2_, deoxynivalenol, T-2 toxin, zearalenone, and ochratoxin A) in TMR were determined as previously described by Li et al. [37]. The health condition of the cows was monitored continuously before and during the entire experiment period.

### 4.2. Sample Collection and Analysis

The cows were milked three times (at 08:30, 15:30, and 21:30) daily, and the individual milk yield was recorded at each milking. Milk samples were collected at days 0, 2, 4, 6, 8, 9, 10, 11, and 12, were proportionally mixed according to the milk yield from each milking, and were stored at −20 °C for the analysis of the milk AFM_1_. Furthermore, one aliquot of the milk samples from day 8 was immediately sent to the Xinjiang Dairy Herd Improvement testing center for an analysis of the milk protein, fat, lactose, milk urea nitrogen (MUN), and somatic cell count (SCC), using a near-infrared reflectance spectroscopy analyzer (Seris300 CombiFOSS; Foss Electric, Hillerød, Denmark). The blood samples were collected from the coccygeal vessel into heparinized evacuated tubes before the morning feeding on day 9, were centrifuged at 2000 rpm for 20 min at 4 °C to prepare the plasma, and were stored at −20 °C for the subsequent analysis. The plasma levels of the total protein (TP); albumin (ALB); and the activities of the enzymes, including aspartate amino transferase (AST), alanine amino transferase (ALT), alkaline phosphatase (ALP), γ-glutamyl transferase (GGT), and lactate dehydrogenase (LDH), were analyzed using an automatic biochemistry autoanalyzer (7160, Hitachi) by the Beijing Sino–UK Institute of Biological Technology (Beijing, China).

### 4.3. Quantification of Aflatoxin M_1_ in Milk

The fluid milk samples were incubated at 37 °C for 10 min, and then centrifuged at 5000 rpm for 15 min to separate the fat. Fifty milliliters of supernatant were collected and applied to an immune-affinity column (Clover Technology Group, Beijing, China) at a steady flow rate of 2 mL min^−1^ using the vacuum system. The column was washed twice with distilled water (10 mL), then eluted with 2 mL of methanol. The eluent was evaporated to dryness at 40 °C under a stream of nitrogen, and reconstituted in 200 μL of distilled water. Chromatographic analyses were performed with an HPLC system (Shimadzu LC-10 AT, Shimadzu, Tokyo, Japan) consisting of a post-column photochemical derivation and a RF-20A fluorescence monitor. The separation of AFM_1_ was achieved by a reverse phase column (DIKMA, C18, 5μm, 15 × 4.6 mm ID). The mobile phase was isocratic acetonitrile:methanol:water (24:8:68, *v*/*v*/*v*), wherein the flow rate was 1 mL min^−1^. The wavelengths for excitation and emission were 365 nm and 435 nm, respectively. The injection volume was 20 μL.

### 4.4. Calculations

AFM_1_ excretion (μg d^−1^) = concentration of AFM_1_ in milk (ng L^−1^) × milk yield (kg d^−1^)

Transfer rate (%) = excretion of AFM_1_ (ng d^−1^)/AFB_1_ consumption (ng d^−1^) × 100%

The SCC values were divided by 1000 and were natural logarithm transformed before analysis.

### 4.5. Statistical Analysis

The milk yield, milk components, and plasma biochemical parameters were analyzed via one-way ANOVA using SAS (version 9.2; SAS Institute, Cary, NC, USA). Duncan’s multiple range tests were conducted when a significant difference was detected among the treatments. Milk AFM_1_ content (concentration, excretion, and transfer rate) at steady-state (days 4 to 8) were analyzed by one-way ANOVA with day of collection as repeated measures. The significance level was set at 0.05.

## Figures and Tables

**Figure 1 toxins-11-00161-f001:**
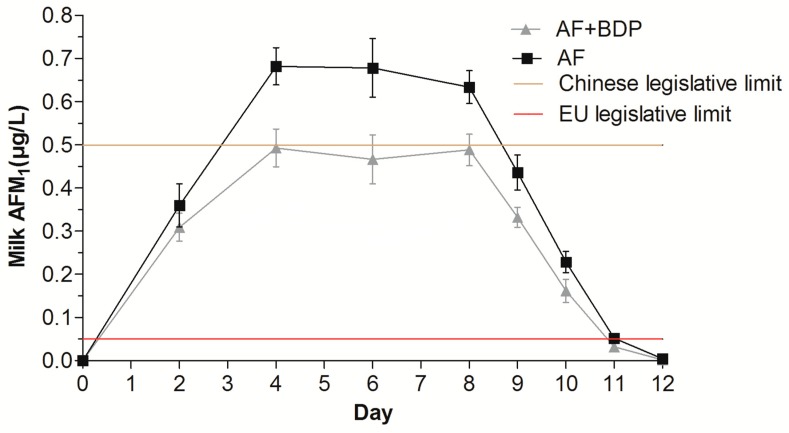
Effect of dietary addition of aflatoxin B_1_ (AFB_1_) with or without *Bacillus subtilis* ANSB060 biodegradation product (BDP) on the concentration of aflatoxin M_1_ (AFM_1_) in the milk of dairy cows. AFB_1_-dosing period = days 1 to 8; clearance period = days 9 to 12. AF: aflatoxin diet, containing control diet (CON), consisting of a basal total mixed ration (TMR) plus 63 μg of AFB_1_ kg^−1^ of diet dry matter; AF + BDP: aflatoxin diet plus BDP, containing AF plus BDP at 0.2% of diet dry matter.

**Table 1 toxins-11-00161-t001:** Effect of the dietary addition of aflatoxin B_1_ (AFB_1_) with or without *Bacillus subtilis* ANSB060 biodegradation product (BDP) on the milk production and composition of dairy cows (*n* = 8).

Item	Dietary Treatment ^1^			SEM	*p*-Value
	CON	AF	AF + BDP		
Milk yield (kg d^−1^)	20.18	19.33	19.32	0.83	0.71
Fat (%)	3.43	3.34	3.37	0.18	0.94
Protein (%)	3.34	3.48	3.31	0.09	0.40
Lactose (%)	4.86	4.75	4.89	0.12	0.69
MUN ^2^ (mg dL^−1^)	11.38	12.07	11.88	0.53	0.65
SCC ^3^ (log cells μL^−1^)	5.37	5.70	5.50	0.26	0.67

^1^ CON: control diet, consisting of a basal total mixed ration (TMR); AF: aflatoxin diet, containing CON plus 63 μg of AFB_1_ kg^−1^ of diet dry matter; AF + BDP: aflatoxin diet plus BDP, containing AF plus BDP at 0.2% of diet dry matter. ^2^ Milk urea nitrogen (MUN). ^3^ Natural logarithmic value of somatic cell count (SCC) μL^−1^ of milk. SEM: standard error of the mean.

**Table 2 toxins-11-00161-t002:** Effect of dietary addition of aflatoxin B_1_ (AFB_1_) with or without *Bacillus subtilis* ANSB060 biodegradation product (BDP) on plasma metabolite parameters of dairy cows (*n* = 8).

Item	Dietary Treatment ^1^			SEM	*p*-Value
	CON	AF	AF + BDP		
TP (g dL^−1^)	7.10	7.41	7.23	0.21	0.57
ALB (g dL^−1^)	3.19	3.19	3.05	0.07	0.34
AST (U L^−1^)	89.63	92.13	83.63	3.09	0.16
ALP (U L^−1^)	33.13	34.13	32.38	1.88	0.81
ALT (U L^−1^)	42.50	43.88	39.75	1.84	0.29
GGT (U L^−1^)	39.38	39.88	37.13	2.89	0.78
LDH (U L^−1^)	945.88 ^b^	1106.75 ^a^	1032.88 ^ab^	36.52	0.02

^a,b^ Values in the same row with no common superscript differ significantly(*p* < 0.05). ^1^ CON: control diet, consisting of a basal total mixed ration (TMR); AF: aflatoxin diet, containing CON plus 63 μg of AFB_1_ kg^−1^ of diet dry matter; AF + BDP: aflatoxin diet plus BDP, containing AF plus BDP at 0.2% of diet dry matter. ALP: alkaline phosphatase; AST: aspartate amino transferase; GGT: γ-glutamyl transferase; TP: total protein; ALB: albumin; LDH: lactate dehydrogenase.

**Table 3 toxins-11-00161-t003:** Effect of the dietary addition of aflatoxin B_1_ (AFB_1_) with or without *Bacillus subtilis* ANSB060 biodegradation product (BDP) on the concentration, excretion, and transfer rate of aflatoxin M_1_ (AFM_1_) in the milk of dairy cows at steady-state (days 4 to 8) (*n* = 8).

Item	Dietary Treatment ^1^		SEM	*p*-Value
	AF	AF + BDP		
AFM_1_ concentration (ng L^−1^)	665 ^a^	483 ^b^	9	<0.01
AFM_1_ excretion ^2^ (μg d^−1^)	12.71 ^a^	9.14 ^b^	1.58	<0.01
Transfer rate ^3^ (%)	1.06 ^a^	0.76 ^b^	0.01	<0.01

^a,b^ Values in the same row with no common superscript differ significantly(*p* < 0.05). ^1^ CON = basal TMR; AF = basal TMR + 63µg kg^−1^ AFB_1_, AF + BDP= basal TMR + 63µg kg^−1^ AFB_1_ + 2g kg^−1^ BDP. ^2^ AFM_1_ excretion (μg d^−1^) = concentration of AFM_1_ in milk (ng L^−1^) × milk yield (kg d^−1^). ^3^ Transfer rate (%) = excretion of AFM_1_ (ng d^−1^)/AFB_1_ consumption (ng d^−1^) × 100.

**Table 4 toxins-11-00161-t004:** Ingredients and chemical composition of the basal diet.

Item	Amount
Ingredient composition (% of DM)	
Corn silage	41.5
Alfalfa silage	14.9
Flaked corn	12.1
Cottonseed meal	4.9
Distillers dried grains with soluble (DDGS)	9.7
Soybean meal (47% CP)	2.5
Corn gluten feed	12.2
Mineral and vitamin mix ^1^	2.2
NEL (Mcal kg^−1^ of DM)	1.51
Chemical composition (% of DM)	
Crude protein (CP)	15.5
Fat	3.5
Neutral detergent fiber(NDF)	42.6
Acid detergent fiber(ADF)	25.5
Starch	17.2
Aflatoxins B_1_, B_2_, G_1_, and G_2_	ND ^2^
Deoxynivalenol	ND ^2^
T-2 toxin	ND ^2^
Zearalenone	ND ^2^
Ochratoxin A	ND ^2^

^1^ The mineral and vitamin mix was formulated with 20% salt, 18% Ca, 10% P, 18 mg kg^−1^ of Co, 600 mg kg^−1^ of Cu, 700 mg kg^−1^ of Mn, 800 mg kg^−1^ of Zn, 16 mg kg^−1^ of I, 16 mg kg^−1^ of Se, 30 mg kg^−1^ of Fe, 810 mg kg^−1^ of niacin, 40 mg kg^−1^ of biotin, 300,000 IU kg^−1^ of vitamin A, 7600 IU kg^−1^ of vitamin D_3_, and 2500 IU kg^−1^ of vitamin E. ^2^ ND: not detected. DM: dry matter. CP: crude protein.

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
