# Peer review of "Efficacy of Bacillus subtilis ANSB060 Biodegradation Product for the Reduction of the Milk Aflatoxin M1 Content of Dairy Cows Exposed to Aflatoxin B1"

_toxins, 2019, doi:10.3390/toxins11030161_

Round 1

Reviewer 1 Report

see attached file

Author Response

Thanks for your comments.

--product or products?? it is not clear what you used for aflatoxin biodegradation. It seems that you used B. subtilis spores. In this case, the title should be something like that: Efficecy of B. subtilis ANS... for reducing AFL content/level in....

BPD is the fermentation product of Bacillus subtilis ANSB060, which not only contains B. subtilis spores, but also aflatoxin-degrading enzymes secreted by B. subtilis ANSB060. BDP is produced with industrial fermentation and dry-processing technologies.

--please, provide level of significance。

A:we added (p<0.05).

--statistic?? no significative (level?) differences..

A: we have changed " no difference" as "no significant difference".

--reference needed: Accinelli, C., Mencarelli, M., Saccà, M.L., Vicari, A., Abbas, H.K.

Managing and monitoring of Aspergillus flavus in corn using bioplastic-based formulations

(2012) Crop Protection, 32, pp. 30-35. 

A:we have added the reference.

--to metabolize/degrade or biodegrade..

A: we have changed "bio-transform" as "biodegrade“.

--B subtilis ANS...

A: we change BDP as Bacillus subtilis ANSB060 biodegradation product here. 

--please, explain the method that you used, etc.

A: for the examination of CFU, plate count method was used according to the

instruction of the ISO 4833: 2003 standard (2003).

--... and what about the detection limit??

A: we add the detection limit 0.005 μg/L and quantification limit 0.01μg/L,respectively. 

Reviewer 2 Report

Interesting manuscript, well written and significant impact on agriculture.  Minor comments were as following:

Line 32: Spell AFM1 here for the first time.

Author Response

Thanks for your comments. 

  In line 29, we change afltatoxin M1 as aflatoxin M1(AFM1).

Reviewer 3 Report

The authors provide an interesting study investigating the efficacy of Bacillus subtilis ANSB060 biodegradation product (BDP) for the reduction of milk aflatoxin M1 content of dairy cows.

The Introduction section is an excellent written part of this manuscript providing important literature background of this topic.

The methodology of the experiment is correct, some minor changes are needed.

-    In Table 1 milk urea nitrogen (MUN) is written twice as BUN (L96, L98). Please correct them.

-    As remark for Table 4, please insert information about the methods used for the mycotoxin analysis of TMR (except the AFB1 content) and also show the detection limits.

-    In Table 2 at L238 use the term “Crude protein (CP)” and add also the NEl content of the TMR.

-    In the list, change the order of Ca and P supplementation (Ca was more, than P).

-    At L215 the result is 1197 ug, instead of 1200 ug.

-    Were the control cows “sham treated” during the aflatoxin-dosing phase as the AF and the AF+BDP animals, receiving “placebo” gelatine capsules 2 times per day with balling gun?

-    At L189 delete the “%” sign after the 100.

-    At L136, please delete (d 9) and put it at the end of the same sentence
L137 … within 24 h (d 9).

Overall, the study is interesting and provides new data for the reduction of milk AFM1 concentration by BDP. I agree with the last sentence of the Conclusions, that in the future more research is needed to clarify the mechanisms of AFB1 detoxification in vivo.

Author Response

Thanks for you comments.

-    In Table 1 milk urea nitrogen (MUN) is written twice as BUN (L96, L98). Please correct them.

A:we have corrected them. please see table 1.

-    As remark for Table 4, please insert information about the methods used for the mycotoxin analysis of TMR (except the AFB1 content) and also show the detection limits.

A: thanks for your suggestion. In fact , our lab have established methods for determining mycotoxins in feeds and feedstuffs with HPLC, as you can see in our previous report[1]. Besides, we add information"The contents of mycotoxins (AFB1, AFB2, AFG1, AFG2, deoxynivalenol, T-2 toxin, zearalenone and ochratoxin A) in TMR were determined as previously described by Li et al." in our revised manuscript.

-    In Table 2 at L238 use the term “Crude protein (CP)” and add also the NEl content of the TMR.

-    In the list, change the order of Ca and P supplementation (Ca was more, than P).

A: we have used the term  “Crude protein (CP)” and added the NEL content of the TMR. And we also change the order of Ca and P supplementation.

-    At L215 the result is 1197 ug, instead of 1200 ug.

A: we have changed 1200 as 1197.
-    Were the control cows “sham treated” during the aflatoxin-dosing phase as the AF and the AF+BDP animals, receiving “placebo” gelatine capsules 2 times per day with balling gun?

A: Yes, cows in control also receive  gelatine capsules containing ground corn twice a day with balling gun.

-    At L189 delete the “%” sign after the 100.

A: we have deleted the “%” sign after the 100.

-    At L136, please delete (d 9) and put it at the end of the same sentence
L137 … within 24 h (d 9).

A: We have put  (d 9) at the end of the sentence.

Reference

[1]  Li, X.; Zhao, L.; Fan, Y.; Jia, Y.; Sun L.; Ma, S.; Ji, C.; Ma, Q.; Zhang, J. Occurrence of mycotoxins in feed ingredients and complete feeds obtained from the Beijing region of China. J. Anim. Sci. Biotechnol. 2014, 37, 2–8.